*Method*

# A solid-phase transfection platform for arrayed CRISPR screens

Özdemirhan Serçin[1] (iD), Sabine Reither[2], Paris Roidos[1] (iD), Nadja Ballin[1] (iD), Spyridon Palikyras[1], Anna Baginska[1], Katrin Rein[1], Maria Llamazares[1] (iD), Aliaksandr Halavatyi[2], Hauke Winter[3,4], Thomas Muley[4,5], Renata Z Jurkowska[1] (iD), Amir Abdollahi[6,7] (iD), Frank T Zenke[8], Beate Neumann[2] & Balca R Mardin[1,*] (iD)

## Abstract

Arrayed CRISPR-based screens emerge as a powerful alternative to pooled screens making it possible to investigate a wide range of cellular phenotypes that are typically not amenable to pooled screens. Here, we describe a solid-phase transfection platform that enables CRISPR-based genetic screens in arrayed format with flexible readouts. We demonstrate efficient gene knockout upon delivery of guide RNAs and Cas9/guide RNA ribonucleoprotein complexes into untransformed and cancer cell lines. In addition, we provide evidence that our platform can be easily adapted to high-throughput screens and we use this approach to study oncogene addiction in tumor cells. Finally demonstrating that the human primary cells can also be edited using this method, we pave the way for rapid testing of potential targeted therapies.

**Keywords** arrayed screens; CRISPR/Cas9; gRNA/RNP delivery; solid phase; transfection platform

**Subject Categories** Chromatin, Transcription & Genomics; Methods & Resources

**Mol Syst Biol. (2019) 15: e8983**

## Introduction

The ongoing development of tools for targeted genome editing using CRISPR (clustered regularly interspaced short palindromic repeat) has revolutionized functional genomics (Shalem *et al*, 2015). To date, CRISPR-based forward genetic screens in pooled format have largely been used to study gene essentiality and synthetic lethality in different contexts (Shalem *et al*, 2014; Hart *et al*, 2015; Wang *et al*, 2015, 2017). These screens are largely restricted to measuring growth phenotypes or to phenotype/cellular markers that can be selected by fluorescence-activated cell sorting (FACS) and require next-generation sequencing (NGS) based readouts to analyze the data. In contrast, using fluorescence, luminescence, or imaging-based assays, arrayed screens can be used to study a wider range of cellular phenotypes such as cellular morphology, cell cycle stage, and protein or mRNA localization (Henser-Brownhill *et al*, 2017; de Groot *et al*, 2018). So far, a number of arrayed screens were described using viral-based transduction methods where each well on a multi-well plate is transduced with a virus that can integrate into the cells (Hultquist *et al*, 2016; McCleland *et al*, 2016; Datlinger *et al*, 2017). This experimentally tedious procedure can introduce potential biases since each viral MOI needs to be calculated to make sure each cell received the same amount of virus and an antibiotic selection is required to enrich for transduced cells. Since viral-based transduction strategies can be laborious for arrayed screens, a good alternative is the use of synthetic gRNA complexes that can be directly transfected into Cas9 protein-expressing cells (Shalem *et al*, 2015). However, classical lipofection-based methods may suffer from high variation between individual experiments and their success largely depends on the cell line. As an alternative, the first idea of spotting nucleic acids onto solid surfaces was implemented by Ziauddin & Sabatini (Ziauddin & Sabatini, 2001) where different cDNAs were spotted onto glass slides. In these slides, eGFP expression measurements led to the discovery of gene functions involved in tyrosine kinase signaling, apoptosis, and cell adhesion. This idea gave rise to further developments such as siRNA microarrays, which were implemented to discover genes involved in cytokinesis, and proteasome-mediated proteolysis (Silva *et al*, 2004) as well as in combination with high-content microscopy revealed mitotic

1   BioMed X Innovation Center, Heidelberg, Germany
2   Advanced Light Microscopy Facility, European Molecular Biology Laboratory, Heidelberg, Germany
3   Department of Surgery, Thoraxklinik at University Hospital Heidelberg, Heidelberg, Germany
4   Translational Lung Research Center (TLRC) Heidelberg, Member of the German Center for Lung Research (DZL), Heidelberg, Germany
5   Thoraxklinik at University Hospital Heidelberg, Heidelberg, Germany
6   Division of Molecular and Translational Radiation Oncology, National Center for Tumor Diseases (NCT), and German Cancer Research Center (DKFZ), Heidelberg University Hospital, Heidelberg, Germany
7   Clinical Cooperation Unit Translational Radiation Oncology, German Cancer Consortium (DKTK) Core Center Heidelberg, Heidelberg, Germany
8   Translational Innovation Platform Oncology, Merck KGaA, Darmstadt, Germany
    *Corresponding author. Tel: +49 6221 426 11 701; E-mail: mardin@bio.mx

phenotypes and regulators of secretory pathway in HeLa cells (Neumann *et al*, 2010; Simpson *et al*, 2012). While being instrumental for the development of high-throughput arrayed screens that allowed the investigation of cellular physiology as well as cytotoxicity and drug screening, microarrays are typically limited to imaging-based readouts and use only a few numbers of cells per spot. Thus, an efficient delivery system applicable to a wide range of cell types with various readouts is missing. Here, inspired by previous methods using cDNA (Ziauddin & Sabatini, 2001) or siRNA (Erfle *et al*, 2008) we developed a solid-phase transfection platform that facilitates arrayed CRISPR screens in multi-well plates enabling low cytotoxicity and high efficiency with various readouts such as microscopy, flow cytometry, and ATP-based viability measurements. We demonstrate successful delivery of synthetic guide RNAs to 20 different Cas9-expressing cell lines. In addition, we establish delivery of Cas9:guide RNA ribonucleoprotein (RNP) complexes to cancer cell lines and we provide evidence that our platform is suitable to study oncogene addiction in tumor cell lines. Finally, we provide evidence that not only cancer cell lines, but also primary cells can be edited by using epithelial cells from lung adeno- and squamous cell carcinoma patients. Taken together, we provide a novel platform for rapid testing of potential targeted therapies.

## Results and Discussion

### Establishment of solid-phase transfection with gRNAs

In solid-phase transfection, the microwell plates are coated with the transfection reagent and the synthetic crRNA:tracrRNA complexes (hereafter referred to as gRNAs) that are stabilized by sucrose and gelatin. Freeze-dried plates with these complexes can either be stored for long periods of time or can directly be used by seeding cells on these pre-coated plates (Fig 1). As a proof-of-

principle, we first established and used our solid-phase transfection platform to target genes with gRNAs in cell lines expressing Cas9. We used *TP53*-deficient RPE-1 cells with doxycycline-inducible Cas9 expression (hereafter referred to as RPE-1) (Fig EV1) and tested four target genes that exhibit clear phenotypes upon disruption that can be followed by microscopy, assessing either the changes in the nuclear morphology; *PLK1*, *CCNA2* or measuring the loss of signal after antibody staining; *GOLGA2* and *MKI67* (Fig 2).

1  Plk1 is a cell cycle kinase with various functions in mitotic spindle formation (Sumara *et al*, 2004). Consistent with this role, RPE-1 cells transfected with a gRNA targeting *PLK1* accumulated in prometaphase already 24 h after transfection (Fig 2A), indicating a cell cycle arrest, followed by cell death after 72 h. Notably, the phenotypic penetrance was similar to *PLK1* knockdown by siRNA (Fig EV2A–C).
2  Cyclin A is known to have dual role in controlling the cell cycle by the activation of cyclin-dependent kinases and has been shown to be involved in initiation of DNA replication and mitotic entry (Pagano *et al*, 1992). We transfected cells with a gRNA targeting *CCNA2* and followed the cells after 72 h of transfection. Consistent with its siRNA knockdown phenotype (Neumann *et al*, 2010), knockdown of CCNA2 by solid-phase transfection of gRNA resulted in a marked increase in the number of cells with large nuclei (Fig 2B and C).
3  Golga2 is a well-established Golgi marker (Nakamura *et al*, 1995). We transfected the gRNAs targeting *GOLGA2* locus (Fig EV2D) and assessed GOLGA2 protein levels by immunofluorescence and observed a marked decrease in Golga2 protein levels 72 h after transfection (Fig 2D and E).
4  Ki67 is a marker of proliferating cells (Gerdes *et al*, 1984). We measured the levels of Ki67 in the nuclei by immunofluorescence as well as by FACS (Figs 2F and EV2E). Upon 72 h of

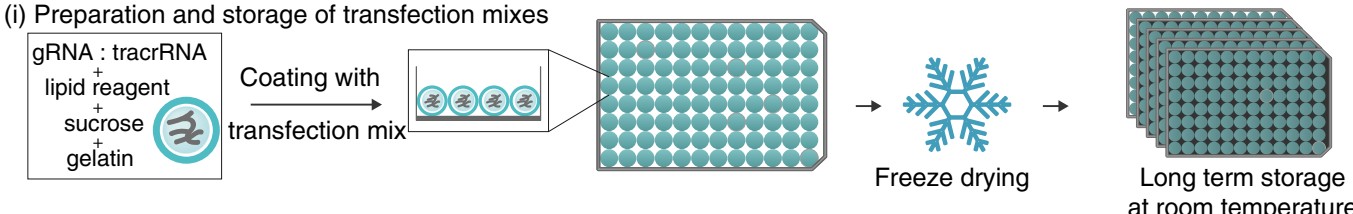

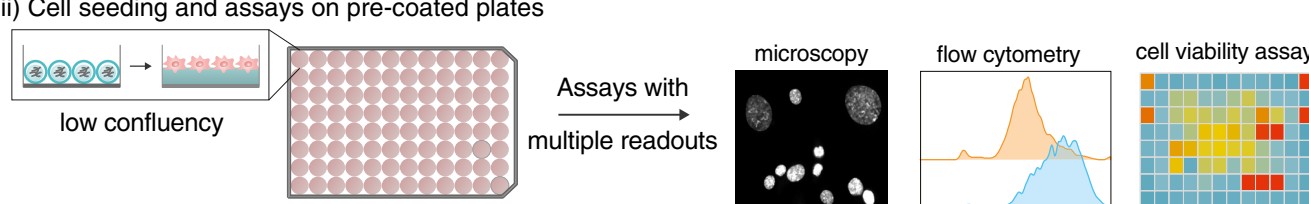

**Figure 1.  Workflow for solid-phase transfection.**

(i) In solid-phase transfection, the microwell plates are coated with the transfection mixes consisting of synthetic gRNAs, lipid reagent, sucrose, and gelatin. The microwell plates are then freeze dried and can either be stored for long periods of time or (ii) the cells can directly be seeded on these pre-coated plates. A wide range of readouts such as microscopy, flow cytometry, or cell viability assays is possible.

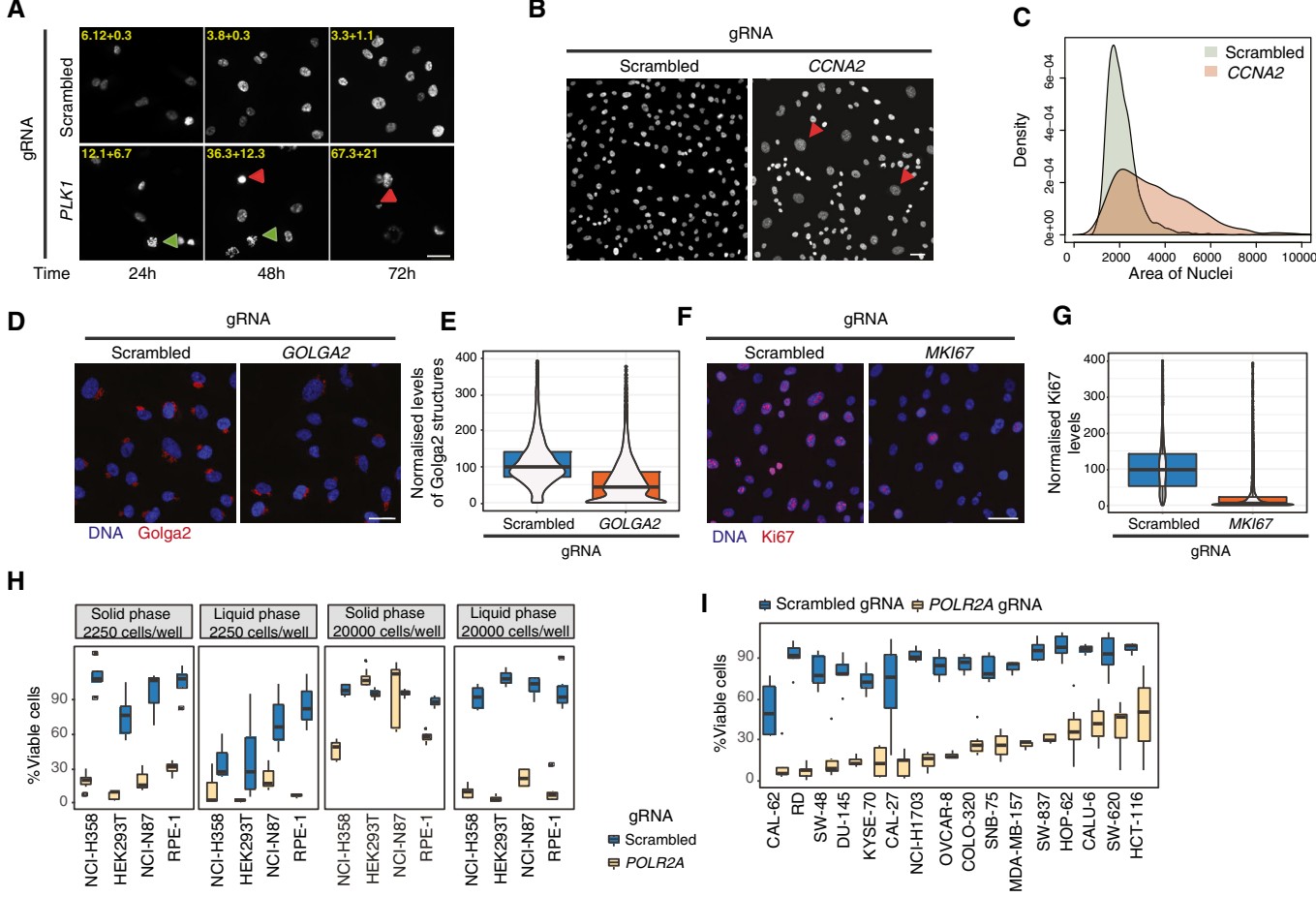

**Figure 2. Solid-phase transfection for delivery of synthetic guide RNAs.**

A   Solid-phase transfection of nontargeting (scrambled) or *PLK1* targeting gRNAs into Cas9-expressing RPE-1$^{TP53-/-}$ cells. Cells were fixed after 24, 48, and 72 h and imaged after DNA staining with Hoechst. Green arrowheads indicate examples of prometaphase-arrested cells, and the red arrowheads indicate examples of dead cells due to Plk1 downregulation. Phenotypic penetrance is indicated at the top of each panel with standard deviation derived from two independent experiments. Scale bar, 20 μm.

B   Solid-phase transfection of nontargeting (scrambled) or *CCNA2* targeting gRNAs into Cas9-expressing RPE-1$^{TP53-/-}$ cells. Cells were fixed after 72 h and imaged after DNA staining with Hoechst. Red arrowheads indicate large nuclei, a typical phenotype observed after CCNA2 depletion. Scale bar, 50 μm.

C   Quantification of nuclear size measurements from (B). Data derived from three independent experiments. *P* value (scrambled versus CCNA2) $< 2^{e-16}$, Kolmogorov–Smirnov test.

D   Solid-phase transfection of nontargeting (scrambled), *GOLGA2* targeting gRNA complexes into Cas9-expressing RPE-1$^{TP53-/-}$ cells. Cells were fixed after 72 h, stained with Golga2 antibody (red), and Hoechst (blue) to mark DNA and imaged. Scale bar, 50 μm.

E   Quantification of experiments in (D). Data derived from three independent experiments. Data is represented as violin plots merged with boxplots that extends from min to max with the probability density. Boxes indicate 25$^{th}$ and 75$^{th}$ percentiles. Black lines indicate median values. *P* value (scrambled versus GOLGA2) $< 2^{e-16}$, Mann–Whitney *U* test.

F   Solid-phase transfection of nontargeting (scrambled), *MKI67* targeting gRNA complexes into Cas9-expressing RPE-1$^{TP53-/-}$ cells. Cells were fixed after 72 h, stained with Ki67 antibody (red), and Hoechst (blue) to mark DNA and imaged. Scale bar, 50 μm.

G   Quantification of experiments in (F). Data derived from three independent experiments. *P* value (scrambled versus MKI67) $< 2^{e-16}$, Mann–Whitney *U* test. Data is represented as violin plots merged with boxplots that extends from min to max with the probability density. Boxes indicate 25$^{th}$ and 75$^{th}$ percentiles. Black lines indicate median values.

H   Comparison of phenotypic penetrance with liquid- and solid-phase transfection. RPE-1, HEK293T, NCI-H358, and NCI-N87 cells were transfected with scrambled or *POLR2A* targeting gRNAs in two different cell numbers (2,250 or 20,000 cells/well). Five days post-transfection, cell viability was measured by CellTiter-Glo. Boxplots represent values from at least three independent experiments containing three technical replicates. In the boxplots, centerlines mark the medians, box limits indicate the 25$^{th}$ and 75$^{th}$ percentiles, and whiskers extend to 5$^{th}$ and 95$^{th}$ percentiles.

I   Cell viability measurements after solid-phase transfection targeting *POLR2A* in a panel of cell lines. Cas9-expressing cell lines were transfected with scrambled and *POLR2A* targeting gRNA, and cell viability was assessed after 5 days. The raw values are background subtracted and normalized to the mock controls. Results are from at least three independent experiments containing three technical replicates. In the boxplots, centerlines mark the medians, box limits indicate the 25$^{th}$ and 75$^{th}$ percentiles, and whiskers extend to 5$^{th}$ and 95$^{th}$ percentiles. For all cell lines, *P* values (scrambled versus POLR2A) $< 0.005$. Mann–Whitney *U* test.

transfection of gRNAs targeting *MKI67* gene, we have observed a very significant reduction in Ki67 levels (Fig 2G). Altogether, these data suggest that solid-phase transfection platform is suitable for screens with phenotypic readouts.

After establishing several markers that can be followed up by imaging-based assays in RPE-1 cells, we wondered whether we can extend and apply this system to additional cell lines and for that we wanted to use a rapid, viability-based readout that is compatible with experiments in higher throughput in several different cell lines. To this end, we examined the outcome of *POLR2A* disruption by a luminescence-based cell viability assay measuring cellular ATP levels. *POLR2A* is an essential gene that encodes a subunit of the RNA polymerase II (Young, 1991). Consistently, its disruption led to an 85% reduction in cell viability compared to a nontargeting (scrambled) gRNA (Fig EV2F). We confirmed the specificity of *POLR2A* targeting first at the DNA level by demonstrating the correct targeting of the *POLR2A* locus (Fig EV2G). Second, we

measured the POLR2A protein levels in the nucleus by immunofluorescence. We observed a time-dependent and significant decrease in the POLR2A protein levels, consistent with the previously reported 40-h half-life of POLR2A (McShane *et al*, 2016) (Fig EV2H and I). These results suggest that solid-phase transfection has the potential to be applied to viability-based assays.

High-throughput arrayed screens require minimal handling after transfection to reduce potential biases. For this reason, an important prerequisite for these screens would be to be able to transfect low number of cells with minimal cytotoxicity. We have observed that with liquid transfection some cell lines such as RPE-1 can be efficiently transfected when low number of cells was seeded (2,500 cells/well) as previously reported in the literature for other cell lines (Tan & Martin, 2016; Strezoska *et al*, 2017; de Groot *et al*, 2018). However, some cell lines such as NCI-H358, NCI-N87, or HEK293T that also express doxycycline-inducible Cas9 (Fig EV1) suffer from increased cytotoxicity and high variability under the same conditions. Since different cell lines may respond differently

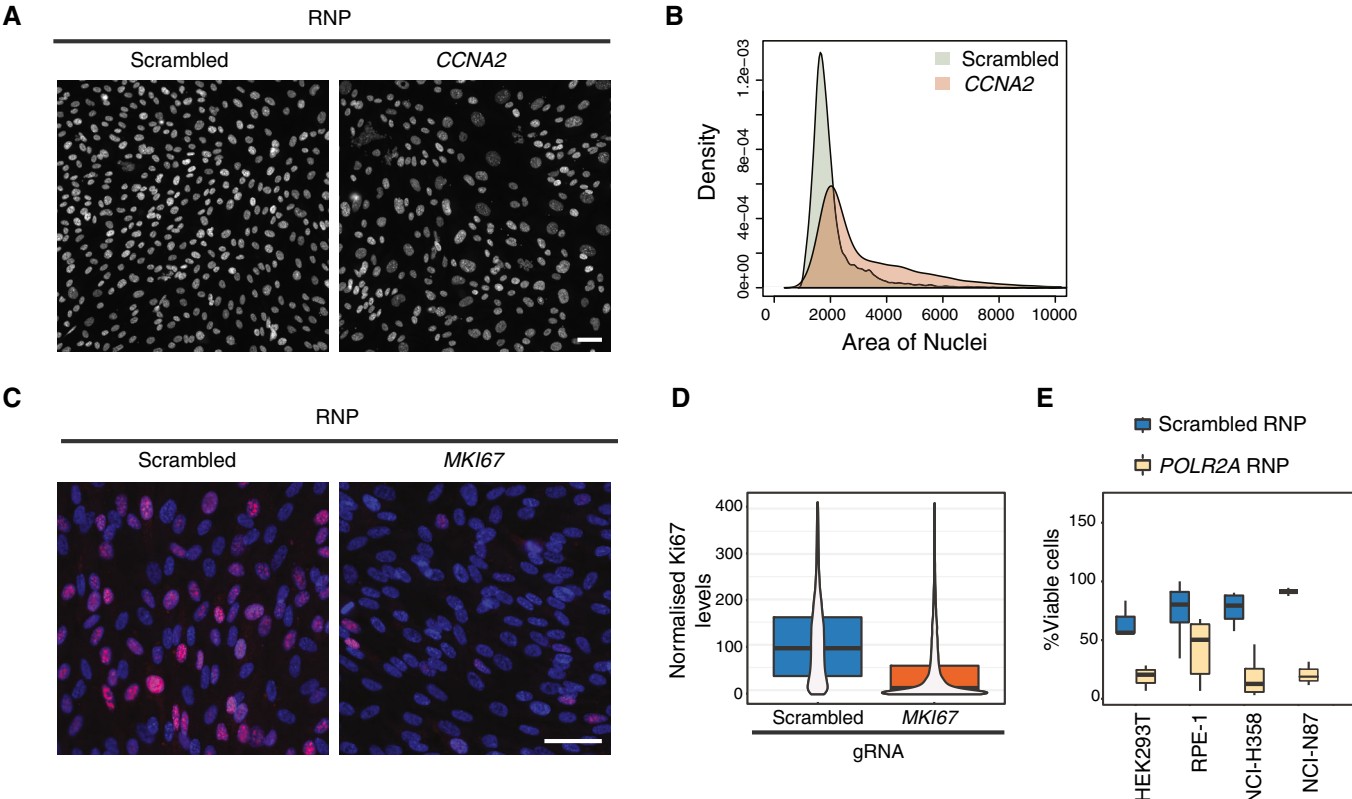

**Figure 3. Solid-phase transfection for delivery of Cas9/guide RNA ribonucleoprotein complexes.**

A  Solid-phase transfection of nontargeting (scrambled) or *CCNA2* targeting RNP complexes into WT RPE-1$^{TP53-/-}$ cells. Cells were fixed after 72 h and imaged after DNA staining with Hoechst. Scale bar, 50 μm.

B  Quantification of experiments in (A). Data derived from two independent experiments. $P$ value (scrambled versus CCNA2) $< 2^{e-16}$, Kolmogorov–Smirnov test.

C  Solid-phase transfection of nontargeting (scrambled), *MKI67* targeting RNP complexes into WT RPE-1$^{TP53-/-}$ cells. Cells were fixed after 72 h, stained with Ki67 antibody (red), and Hoechst (blue) to mark DNA and imaged. Scale bar, 50 μm.

D  Quantification of experiments in (C). Data derived from two independent experiments. Data is represented as violin plots merged with boxplots that extends from min to max together with the probability density. Boxes indicate 25$^{th}$ and 75$^{th}$ percentiles. Black lines indicate median values. $P$ value (scrambled versus MKI67) $< 2^{e-16}$, Mann–Whitney $U$ test.

E  Four cell lines were transfected with RNP complexes with scrambled or *POLR2A* targeting gRNA. 5 days post-transfection, cell viability in each well was measured. Results are from at least three independent experiments containing three technical replicates. In the boxplots, centerlines mark the medians, box limits indicate the 25$^{th}$ and 75$^{th}$ percentiles, and whiskers extend to 5$^{th}$ and 95$^{th}$ percentiles. For all cell lines, $P$ values (scrambled versus POLR2A) $< 0.005$. Mann–Whitney $U$ test.

to changing transfection conditions, we next tested whether solid-phase transfection can be used as a universal strategy to perform arrayed screens in multiple cell lines. Using solid-phase transfection in these cell lines, we observed 70–90% cell death when targeting *POLR2A* with little effect on cell fitness with a nontargeting scrambled gRNA (Fig EV3A). In addition, we showed that the general fitness of the cells remained unaffected even after storage of the lyophilized plates at room temperature or using another set of control gRNAs that target either nonessential genes or intergenic regions (Fig EV3B and C). Next, we compared solid-phase transfection to commonly used methods of liquid-phase transfection. In all tested cell lines, our solid transfection protocol performed best when low number of cells (2,250 cells/well) was seeded. In this case, the transfection efficiencies were similar to that of liquid transfection with high number of cells (20,000 cells/well); however, these high numbers are incompatible with prolonged culturing of cells for downstream applications (Fig 2H). These data suggest that when low number of cells is seeded, the solid-phase transfection allows for the arrayed CRISPR screens to be carried out in an efficient manner for extended periods of time without the need of additional handling steps.

To further validate these results in other cell models, we next generated and tested 16 additional Cas9-expressing cell lines (Fig EV1 and Table EV1) and assessed the effect of targeting *POLR2A* on cell viability. In all these cell lines, we repeatedly observed phenotypic penetrance above 50%, with 10 cell lines reaching median efficiencies above 70% while showing minimal cytotoxicity after scrambled gRNA transfection (Figs 2I and EV3D). In conclusion, our platform is suitable for targeted gRNA screens in a wide range of Cas9-expressing cell lines.

## RNP complexes can be used in solid-phase transfection

Stable inducible Cas9 expression in cells might not always be available, and exogenous expression of Cas9 can be disadvantageous in certain settings. Thus, we tested the compatibility of our platform with RNP complex transfection. To this end, using previously established markers that can be followed by imaging-based assays, we transfected RPE-1 cells with RNP complexes targeting CCNA2 and MKI67 to assess changes in nuclear morphology and reduction in protein levels, respectively. In the case of CCNA2, consistent with the phenotype of the gRNA transfection, we observed an increased rate of cells with large nuclei at 72 h post-transfection (Fig 3A and B). For Ki67, we also observed a marked reduction in the protein levels by immunofluorescence suggesting that solid-phase transfection can deliver RNP complexes that can be assessed by phenotypic readouts (Fig 3C and D). In addition, we tested the efficiency of *POLR2A* targeting using RNPs. In wild-type HEK293T, NCI-N87, and NCI-H358 cells, we observed up to 90% loss in cell viability upon transfection of RNP complexes targeting *POLR2A* (Fig 3E), demonstrating the ability of solid-phase transfection platform to deliver RNPs to several cancer cell lines.

## Solid-phase transfection can be easily adapted to high-throughput screens

The ability to transfect a broad range of cell types with minimal cytotoxicity allowed us to directly adapt the solid-phase platform to high-throughput screens. For this, we designed a multi-well plate where we individually targeted a panel of known oncogenes or kinases that are reported to be misregulated in human tumors along with

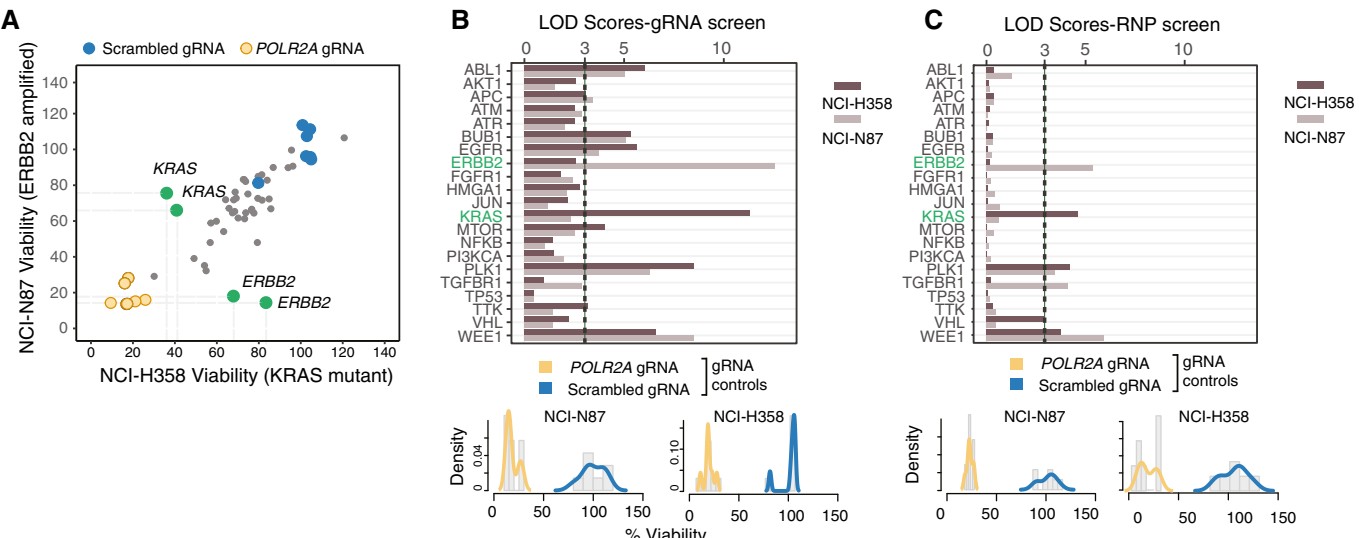

**Figure 4. Arrayed CRISPR screens using solid-phase transfection.**

A Viability of NCI-N87 and NCI-H358 cells upon transfection of 45 different gRNAs by solid-phase transfection. Cas9-expressing NCI-N87 and NCI-H358 cells were seeded on pre-coated plates. Five days post-transfection, cell viability in each well was measured by CellTiter-Glo. Values were background subtracted, normalized to scrambled controls, and plotted against each other. Blue dots represent scrambled controls, whereas the yellow dots represent *POLR2A* gRNAs. Green dots represent the genes that affect the viability in a cell line-dependent manner. Results are representative of three independent experiments.

B, C Logarithm of the odds (LOD) scores of cancer-associated gene KOs in NCI-N87 and NCI-H358 cells derived from screens with gRNAs (B) and RNPs (C). Bottom panels show distribution of positive (*POLR2A*) and negative (scrambled) controls for each experiment. LOD score of 3 is highlighted with a dashed line, indicating statistical significance.

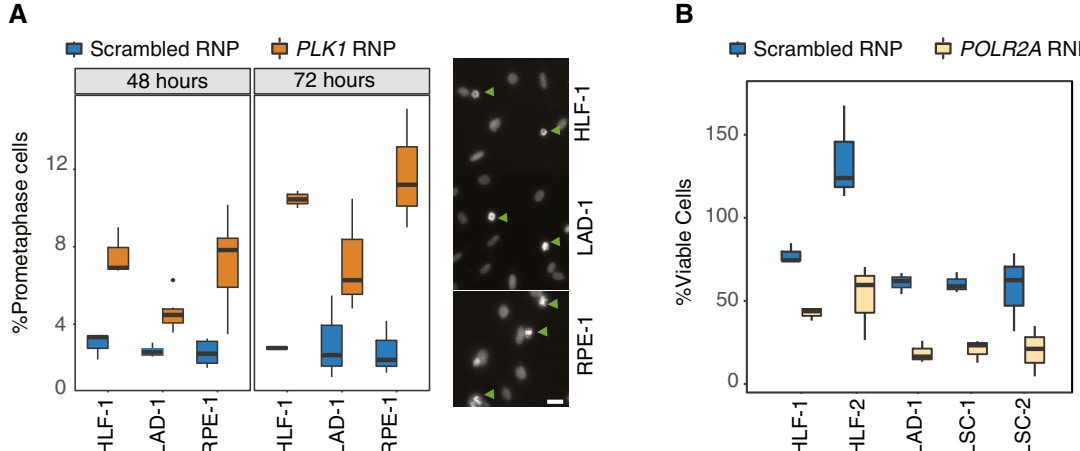

**Figure 5. Solid-phase transfection in primary cells derived from human tumors.**

A   Solid-phase transfection of nontargeting (scrambled) or *PLK1* targeting RNP complexes into human primary lung fibroblasts (HLF-1), primary lung adenocarcinoma cells (LAD-1), or RPE-1$^{TP53-/-}$ cells. 72 hours post-transfection, cells were stained with Hoechst and imaged. Boxplots represent values from three independent experiments containing three technical replicates. For all cell lines, *P* values (scrambled versus Plk1) < 0.05. In the right panels, representative images for each cell line are shown after RNP transfection targeting *PLK1*. Arrowheads indicate the cells that are arrested in prometaphase due to Plk1 downregulation. Scale bars indicate 20 μm. In the boxplots, centerlines mark the medians, box limits indicate the 25th and 75th percentiles, and whiskers extend to 5th and 95th percentiles. Mann–Whitney *U* test.

B   Two human primary lung fibroblasts (HLF) and three primary tumor cell lines derived from patients suffering from either lung squamous carcinoma (LSC) or lung adenocarcinoma (LAD) were transfected with RNP complexes with scrambled or *POLR2A* targeting gRNA. Five days post-transfection, cell viability in each well was measured. Results are from at least three independent experiments containing three technical replicates. In the boxplots, centerlines mark the medians, box limits indicate the 25th and 75th percentiles, and whiskers extend to 5th and 95th percentiles. For all cell lines, *P* values (scrambled versus POLR2A) < 0.005. Mann–Whitney *U* test.

scrambled and *POLR2A* control gRNAs (Table EV2) (Higgins *et al*, 2007; Davoli *et al*, 2013). With this target panel, we tested the potential oncogene addiction of two Cas9-expressing cell lines, namely NCI-H358 and NCI-N87 cells. In both cell lines, *POLR2A* gRNAs caused a significant decrease in cell viability, whereas transfection of the scrambled gRNAs had minimal effects, indicating functional positive and negative controls in our assay system. NCI-H358 cells carry a constitutive activating mutation in the *KRAS*$^{G12C}$ gene, whereas NCI-N87 have a copy number amplification of the *ERBB2* gene (Tate *et al*, 2019). Thus, we predicted that these cell lines would show survival dependency on *KRAS* and *ERBB2*, respectively (Pagliarini *et al*, 2015) (Fig 4A). For the majority of target genes in our panel, the effect of individual gRNAs on viability was similar in both of the cell lines. Strikingly, we observed loss of cell viability upon *KRAS* and *ERBB2* disruption in NCI-H358 and NCI-N87 cell lines, respectively (Figs 4A and EV4). Although it is now acknowledged that high copy numbers affect the viability of the cells when targeting these regions by gRNAs and ERBB2 dependency may be confounded by this effect (Aguirre *et al*, 2016; Munoz *et al*, 2016), the dependency on KRAS does stem from an activating mutation, thus is directly a measure of the oncogenic activity of KRAS that the cells rely on. These data suggest that our system is suitable to detect specific dependencies arising from genomic alterations in cancer cells.

To normalize screening results from different cell lines and to correct for the effect of individual gRNAs, we developed a scoring system based on the logarithm of the odds (LOD), which is a statistical test often used for linkage analysis in populations (Morton, 1955) (Fig EV5A–C). In our case, the LOD score compares the likelihood of the phenotype of a gene KO to be similar to that of POLR2A (our positive control that causes lethality when knocked out) over the likelihood that the phenotype of the gene KO is similar to our negative controls (scrambled gRNA) based on the assumption of

normally distributed control values. A LOD score of 3 means that the likelihood of a gene KO behaving similarly to POLR2A is 1,000 times higher than the likelihood of a gene KO behaving similar to the scrambled control; thus, a high LOD score is an indication of a gene KO behaving similarly to *POLR2A* KO, that is causing loss of cell viability. LOD calculation revealed that genes affecting mitotic division, such as *WEE1* and *PLK1* (Neumann *et al*, 2010), had LOD scores of > 5 in both cell lines (Fig 4B). In addition, we observed *KRAS* and *ERBB2* had the highest LOD scores in NCI-H358 and NCI-N87 cells, respectively. Thus, we conclude that our platform allows prediction of genotype-specific lethalities in cancer cells due to oncogene addiction.

We next asked whether we could obtain comparable results also with solid-phase RNP transfection. For this, we used NCI-H358 and NCI-N87 cells that do not stably express Cas9. Indeed, similar to gRNA transfection, RNP transfection produced high LOD scores for key cell cycle regulators *WEE1* and *PLK1* in both cell lines. Furthermore, *ERBB2* LOD score was among the highest in NCI-N87 cells and *KRAS* LOD score was the highest in NCI-H358 cells (Figs 4C and EV4). These data are consistent with the experiments performed in Cas9-expressing cells transfecting the gRNA complexes, thus showing that solid-phase RNP transfection is an alternative and efficient approach for arrayed CRISPR/Cas9 screens.

### Primary cells can be edited by solid-phase transfection of RNP complexes

One potential application of our platform can be in assessment of targeted therapies. Solid-phase transfection can be used as a rapid way to find the vulnerabilities of certain tumors, if the cells derived from patient samples can also be efficiently transfected. In order to pave the way of using this methodology in targeted therapy, we

tested whether we can also target primary cells by solid-phase transfection of RNP complexes. To this end, we transfected *PLK1* targeting RNP complexes into human primary lung fibroblasts derived from lung parenchyma as well as primary lung adenocarcinoma cells and compared these results to RPE-1 cells. In both RPE-1 cell line and the primary cells, we observed an increased rate of mitotic arrest at 48 h post-transfection (Fig 5A). In addition, we transfected *POLR2A* targeting RNP complexes to two human primary lung fibroblasts and three primary tumor cell lines derived from patients suffering from either lung squamous carcinoma or lung adenocarcinoma. Although in some of these primary cells we observed slightly higher levels of cytotoxicity due to transfection, the loss of viability due to transfection of *POLR2A* targeting RNPs was consistently and significantly higher (Fig 5B). Our results demonstrate that solid-phase transfection platform can be compatible with RNP delivery not only in cancer cell lines but also in human primary cancer cells.

In summary, we developed a solid-phase reverse transfection method that allows efficient delivery of either synthetic gRNAs or Cas9 containing RNP complexes. Our system has several advantages over previous delivery approaches for screening purposes (Tan & Martin, 2016; Bulkescher *et al*, 2017; Strezoska *et al*, 2017). First, the transfection complexes can be easily and efficiently coated onto the plates, and ready-to-use plates allow flexibility in designing experiments. Second, multiple plates can be produced from one transfection mix, eliminating potential batch effects across different experiments. Finally, a wide range of cell lines can be directly seeded on coated plates which allow several downstream applications such as viability measurements or microscopy-based readouts. This reduces potential biases caused by additional handling steps and allows easy automation.

However, we also note couple of limitations in our system. For instance, solid-phase transfection requires additional equipment for freeze drying the transfection mixes and it can only be applied to adherent cells. In addition, due to the time required for the observation of certain phenotypes, solid-phase transfection of gRNAs may be limited to gene products that have fast turnover rates. Arrayed screens need to be completed in a shorter time frame than that of the pooled screens that allow cells to be kept in culture longer time. Currently, with a maximum time frame of 5 days post-transfection, only proteins with fast turnover can be efficiently targeted. To estimate how many proteins can be efficiently targeted by CRISPR/Cas9-based methods such as solid-phase transfection, we utilized the recent study measuring the protein turnover rates in RPE-1 cells. Assuming most of the experiments should be carried out in 3–5 days post-transfection, we predict that at least in RPE-1 cells phenotypes of approximately 17.7% of the proteins may not be efficiently visualized due to their long half-lives (> 120 h) (Fig EV5D). For this reason, we note that a careful assessment of protein turnover rates should be included in experimental design.

Targeted therapy is a powerful concept based on vulnerabilities or addictions of tumors. Although genome, exome sequencing, and expression analysis of oncogenes provide detailed information about the patients, methods that can experimentally test these predictions in a systematic manner are not well-established. Using our solid-phase CRISPR screening platform, we demonstrate efficient delivery of RNP complexes in untransformed cells, two cancer cell lines, human primary lung fibroblasts, and primary tumor epithelial cells. With high flexibility regarding targets and assays, our platform can contribute to future CRISPR-based biomedical applications.

# Materials and Methods

### Reagents and Tools table

| Reagent/resource | Reference or source | Identifier or catalog number |
|---|---|---|
| **Experimental models** | | |
| List of cell lines | | Table EV1 |
| **Synthetic RNA and Recombinant Cas9 protein** | | |
| A list of synthetic RNA used in this study was detailed in Table EV1 | IDT | N/A |
| *Sp.* Cas9 tracrRNA | IDT | 1072533 |
| **Antibodies** | | |
| Anti-POLR2A (Rabbit) | Sigma | HPA021503 |
| Anti-GM130 (GOLGA2) (Mouse) | BD | 610822 |
| Anti-Cas9 | Cell Signaling | CST #14697 |
| Anti-GAPDH | Cell Signaling | CST #5174 |
| Anti-Ki67 | eBioScience | 14-5698-82 |
| Anti-CyclinA | Santa Cruz | sc-751 |
| Anti-Rabbit AlexaFlour-488 Secondary Antibody | Thermo Fisher | A21206 |
| Anti-Mouse AlexaFlour-568 Secondary Antibody | Thermo Fisher | A11004 |
| Anti-Mouse AlexaFlour-488 Secondary Antibody | Thermo Fisher | A11034 |

**Reagents and Tools table**  (continued)

| Reagent/resource | Reference or source | Identifier or catalog number |
|---|---|---|
| **Virus particles** | | |
| hEF1α-TurboGFP-Cas9 Nuclease | Dharmacon | VCAS11864 |
| hCMV-TurboGFP-Cas9 Nuclease | Dharmacon | VCAS11868 |
| **Oligonucleotides and sequence-based reagents** | | |
| Golga2_5 | PCR primer (IDT) | CACTTGCTTGGGTTTCCTCC |
| Golga2_6 | PCR primer (IDT) | AGCAACCACACACAAAAGCA |
| Plk1_5 | PCR primer (IDT) | AGAGAAACCCACCAAGACCC |
| Plk1_6 | PCR primer (IDT) | TGACTTGTGGGTTGTCTCCT |
| Polr2a_3 | PCR primer (IDT) | CAAACTGCCGTAACCTCTGC |
| Polr2a_4 | PCR primer (IDT) | GACTCCCTAGGATTCGTCGG |
| pCW-Cas9 | AddGene | #50661 |
| psPAX2 | AddGene | #12260 |
| pMD2.G | AddGene | #12259 |
| **Chemicals, enzymes and other reagents** | | |
| Recombinant *Sp.* Cas9 protein | IDT | 1081059 |
| Lipofectamine RNAimax | Thermo Fisher | 13778075 |
| Lipofectamine 2000 | Thermo Fisher | 11668-019 |
| Gelatin | Sigma | G3931 |
| Sucrose | Roth | 4621-1 |
| RNA and DNase free water | Thermo Fisher | 4387936 |
| Q5 Hot Start High Fidelity 2X Master Mix | New England Biolabs | M0494 |
| Surveyor mutation detection kit | IDT | 706020 |
| CellTiter-Glo | Promega | G7573 |
| Doxycycline Hydrochloride | Sigma | D3072 |
| Pre-cast Polyacrylamide Gel | Mini-Protean TGX Bio-Rad | |
| RIPA Buffer | Cell Signaling | CST-9806S |
| Sytox Blue | Molecular Probes | S34857 |
| Opti-MEM | Thermo Fisher | 31985-062 |
| Hexadimethrine Bromide | Sigma | H9268-5G |
| Bovine Serum Albumin | Roth | 3737.3 |
| Triton X100 | Sigma | X100-500 |
| Fetal Bovine Serum | Gibco | 10270106 |
| ACK Lysis Buffer | Sigma Aldrich | A1049201 |
| Penicilin/Streptomycin (PS) | Fisher Scientific | 15140122 |
| Amphotericin B | Fisher Scientific | 15290026 |
| HBSS | Fisher Scientific | 14175053 |
| HS DNA Kit | Agilent | 5067-4626 |
| QuickExtract | Epicentre | QE09050 |
| **Software** | | |
| R-base | https://www.r-project.org | 3.5.0 |
| FiJi/ImageJ | https://fiji.sc/ | 2.0/1.2 |
| ZenLite Microscope Software | https://www.zeiss.com | 3.0 |
| Cell Profiler | https://cellprofiler.org/ | 2.2.1 |
| KNIME | https://www.knime.com | 4.0.2 |
| ShinyHTM | https://github.com/embl-cba/shinyHTM | |

**Reagents and Tools table**   (continued)

| Reagent/resource | Reference or source | Identifier or catalog number |
|---|---|---|
| **Culture and coating plates** | | |
| Flat Bottom Black 96-well plates | Zellkontakt | 655090 |
| Flat Bottom Glass Black 96-well plates | Cellvis | P96-1.5H-N |
| Flat Bottom White 96-well plates | Costar | 3903 |
| Flat Bottom transparent 96-well plates | Eppendorf | 0030730119 |
| **Equipment** | | |
| Vacuum Centrifuge | MiVac | QUC-23050-B00 |
| FACSAria II | Becton Dickinson | |
| Glomax Multi+ | Promega | |
| Western Blot Transfer | Bio-Rad | Trans-Blot Turbo |
| Odyssey System | LI-COR Biosciences | |
| BioAnalyzer | Agilent | 2100 |
| Cell/Particle Counter | Beckman-Coulter Z2 | |
| Multidrop Combi Microplate dispenser | Thermo Fisher | 836 |
| Tumor Dissociation Kit | Miltenyi Biotec | 130-095-929 |
| Evos FL Microscope | Thermo Fisher | |
| Zeiss Ax10 Observer D1 fluorescence microscope | Zeiss | |
| Automated Epifluorescent Microscope | Nikon | Ti-E |
| **Other** | | |
| Silica Desiccant Beads | NeoLab | 1-7301 |
| Cell Strainer | BD Falcon | 352350 |

## Methods and Protocols

### Cell lines and treatments

A table detailing cell line media, supplements, and source information can be found in the Table EV1.

### Generation of Cas9-expressing cell lines

Stable and/or inducible expression of Cas9 in cancer cell lines was achieved by lentiviral transduction of Cas9-encoding plasmids. For generating the viruses for the doxycycline-inducible expression of Cas9, HEK293T cells were co-transfected with pCW-Cas9, psPAX2, and pMD2.G plasmids (Addgene). 72 hours post-transfection, supernatant of HEK293T cells was collected and filtered using 44 μm sterile filter. The relevant cell lines were infected with the 50 μl virus containing supernatant and 8 μg/ml Hexadimethrine Bromide (Sigma). Inducible Cas9-expressing cells were selected with 8 μg/ml puromycin until uninfected cells are eliminated. Cas9 expression was assessed in puromycin-resistant clones by immunofluorescence upon 1 μg/ml doxycycline induction. NCI-H1703, OVCAR-8, and RD cell lines were infected with Cas9 lentiviral particles from Dharmacon (VCAS11868), and MDA-MB-157 cell line was infected with Dharmacon (VCAS11864) with a multiplicity of infection of 0.1. GFP-positive cells were sorted twice to achieve maximum purity.

### Liquid-phase transfection

Exponentially proliferating cells at 70–80% confluency were transfected with 2.5 pmol gRNA complexes and 0.3 μl of Lipofectamine RNAiMax (Invitrogen 13778150) according to the manufacturer's protocols. Culture media were replaced 12 h after transfection mixture was added. Transfected or control group cells were split 1:10 into a new flat bottom 96-well plate (Costar 3903) on the second day following transfection.

### Preparation of CRISPR transfection mixes using solid-phase transfection

#### Annealing of gRNAs

In order to prepare 3.3 μM gRNA complexes, 1 μl from the stock solutions of 100 μM cRNA and tracrRNA was added to 28 μls of duplexing buffer (IDT), heated at 95°C for 95 for 5 min. After the incubation, the tubes were left at room temperature for approximately 20–30 min to cool down.

#### For each reaction

1   To achieve 2.5 pmol gRNA complexes in each coated well, 3 μl Opti-MEM/sucrose solution (1.37%w/v) was mixed carefully eight times with 1.75 μl Lipofectamine 2000 or RNaimax (Thermo Fisher).

2   6.75 μl of 3.3 μM gRNA mixture was added and carefully mixed eight times, and the final transfection mix was incubated for 20 min at room temperature.

3   To prepare the gelatin solution, 0.2 g gelatin was dissolved in 100 ml $H_2O$ and heated up to 56°C.

4   After incubation of gRNAs with the lipofectamine/Opti-MEM/Sucrose mix, 7 μl of gelatin (0.2% w/v in water) was added and carefully mixed 8 times. The final mixture was diluted in RNA and DNase-free water (Thermo Fisher) 1:25. Dilution

needs to be done in two steps. The initial dilution step should not exceed 1/10 of the volume of the transfection mix (18 μl transfection mix was diluted with 132 μl $H_2O$ and mixed eight times, and then, 350 μl of $H_2O$ was added and again mixed eight times).

5  From this mix, 50 μl to each well of a 96-well plate was plated.

6  Plates were dried using a MiVac vacuum centrifuge (QUC-23050-B00) accommodating multi-well plates. The MiVac was preheated to 60°C and was set to 37°C immediately after starting the device.

7  For 384-well plates, we prepared the same mixtures and plated 15 μl of the final mix to each well of a 384-well plate.

### Coating of RNP complexes

For coating of ribonucleoprotein complexes, we mixed 3.3 μM gRNA: tracrRNA and 3 μM Cas9 protein (IDT) (diluted in Opti-MEM (GIBCO)) in a 1:1 ratio) and incubated this mixture for 5 min at room temperature.

RNP complexes were then mixed and incubated with Opti-MEM/sucrose solution and Lipofectamine RNAimax (Thermo Fisher) as described above.

### Preparation of CRISPR transfection mixes using solid-phase transfection (96 wells—protocol using Liquidator)

1  2835 μl duplexing buffer was mixed with 105 μl tracrRNA (from 100 μM stock concentration), and from this mix, 28 μl was distributed to each well of a 96-well PCR plate with a 12-channel pipette.

2  2 μl of crRNA complexes was added from the crRNA master plate to the PCR plate with tracrRNA and duplex buffer.

3  The gRNA mixture was incubated at 95°C for 5 min (PCR machine).

4  The PCR plate was removed and left on the bench for app 10–20 min to cool down to room temperature.

5  For a full 96-well plate, 510 μl Opti-MEM/sucrose solution (1.37% w/v) and 298 μl Lipofectamine 2000 (or Lipofectamine RNAimax for RNP) were mixed.

6  7 μl from this mixture was pipetted into each well of the 96-well plate, and the plate was centrifuged shortly and carefully up to 500 rpm.

7  15–30 μl of 0.2% gelatin was pipetted into each well of the 96-well plate, and the plate was centrifuged shortly and carefully up to 500 rpm.

8  6.5 μl of gRNA complexes with a concentration of 3.3 μM was transferred into an empty 96-well plate.

9  5 μl of the transfection reagent source plate containing Opti-MEM/Sucrose and Lipofectamine 2000 was added to the plate containing gRNA complexes and mixed eight times. The plate was centrifuged shortly and carefully up to 500 rpm.

10  Transfection mixes were incubated at room temperature for 20 min.

11  After incubation, 7 μl of gelatin was added and mixed 8 times.

12  81.5 μl nuclease-free $H_2O$ was added into a 96-deep-well plate.

13  The transfection mix was added into the 96-deep-well plate and mixed 8 times with a volume of 50 μl.

14  A total of 350 μl nuclease-free $H_2O$ (175 μl pipetted twice) was added to the 96-deep-well plate and mixed with a volume of 200 μl volume.

15  50 μl of these diluted transfection mixes was transferred into 8 empty 96-well plates.

16  The plates were immediately dried in speed-vac (MiVac). When drying 8 plates, the following settings were used: The speed-vac was preheated to 60°C, and then, the temperature was changed to 37°C immediately after starting the device. (Note: It is important to not leave the finished plates outside for too long, as this may affect the success of the coating process).

17  The ready plates were stored in sealed boxes with silica desiccant (drying pearls) for future use. We note that while the gRNA plates can be stored over long periods of time in these sealed boxes, the time of storage of RNP complexes should be tested for each cell line as in some cases we observed loss of fitness after 1 week of storage at room temperature.

### Surveyor assay

After medium removal from the cells, lysis was performed directly in 96-well plates according to the manufacturer's protocol using QuickExtract lysis reagent (Epicentre). The cell lysates of the triplicates were collected in one single tube. PCR amplification was done, using Q5 Hot Start High-Fidelity 2X Master Mix (#M0494, New England Biolabs) according to manufacturer's protocol. The enzyme digest of mispaired dsRNA was done using Surveyor endonuclease (#M0302, New England Biolabs) according to the manufacturer's protocol. The DNA was loaded on bioanalyzer using HS DNA kit.

### Viability measurements

For measurements of cell viability, we measured total ATP levels in 96-well plates using CellTiter-Glo Luminescent Viability Assay (G7573-Promega). An equal volume of DMEM and CellTiter-Glo (25 μl) was mixed at room temperature immediately before adding the mix to 96 wells containing adherent cells. Growth media were fully removed before the addition of 50 μl of CellTiter-Glo mix. Cells were incubated in a dark environment for 10 min at room temperature before reading of luminescence using Instinct software with factory preinstalled settings using GloMax Multi+. Data are exported to R for further processing. Relative viability values of control (scrambled) or gRNA transfected were calculated by dividing individual raw luminescence measurements to average of no-transfection (mock) control.

### Immunoblotting

Cells HEK293T and RPE-1 are treated with doxycycline (1 μg/ml) for 24 h. Whole cell extracts of HEK293T and RPE-1 were generated with RIPA buffer (CST—9806S). Equal amounts of protein (25 μg/ml) were loaded on a 7.5% precast polyacrylamide gel (Mini-PROTEAN® TGX™, Bio-Rad). The proteins were then transferred to a nitrocellulose membrane (Trans-Blot® Turbo™, Bio-Rad) using a transfer apparatus according to the manufacturer's protocols (Bio-Rad). After incubation with 10% nonfat milk in TBS-T (10 mM Tris, pH 8.0, 150 mM NaCl, 0.5% Tween 20) for 30 min, the membrane was washed three times with TBS-T and incubated with antibodies against Cas9 (1:1,000, CST #14697) and Gapdh (1:10,000, CST #5174) at 4°C for 12 h. Membranes were washed three times and incubated with 1:10,000 dilution of IRDye 680RD and IRDye 800CW secondary antibodies for 2 h. Blots were washed with TBS-T three times and developed with the Odyssey system (LI-COR Biosciences).

                                                                        

### Cell seeding, fixation, and immunofluorescence

Cells were trypsinized and were passed through a 70-µm cell strainer to achieve a single cell suspension, (BD Falcon, 352350). Cells were counted with a Beckman coulter Z2 particle counter with 10–20 µm setting, and for 96 wells, we seeded 2,250 cells/well using an automated cell seeder (Thermo Fisher Multidrop combi Microplate Dispenser type 836). For transfection of RNP complexes, an additional step was used where the cells were shortly centrifuged at 200× *g* to remove residual trypsin.

Cells stained with Polr2a antibody were seeded in flat bottom 96-well plates (Zellkontakt, 655090). Cells were fixed with 4% PFA for 10 min, washed with PBS, and then blocked with 10% FBS—0.1% Triton X-100 for 1 h. Cells were then stained with primary antibody Anti-Polr2a (1:1,000, Rabbit, Sigma HPA021503) and were incubated in 3% BSA overnight at four degrees. Cells were then washed 3× with PBS and then stained with secondary antibody Alexa-488 (Anti-Rabbit 1:800) and Hoechst (1:4,000) for 1 h at 37 degrees. Cells were then washed 3× with PBS and then stored in 4 degrees in the dark until observed under the microscope.

Cells stained with GOLGA2 and Ki67 antibodies were seeded in 96-well glass bottom plates (Cellvis, P96-1.5H-N), fixed for 12 min with 4% PFA at room temperature, washed three times with D-PBS, and permeabilized for 12 min at room temperature with 0.5% Triton X-100 and 1:5,000 Hoechst33342 in D-PBS. Cells were then stained for 1 h at room temperature (BD, mouse anti GM130 (GOLGA2), #610822, 1:400) or overnight at 4 degrees (Cell Signaling Technologies rabbit-anti-Ki67, #D3B5, 1:400) with primary antibody in 0.02% Triton X-100 in D-PBS, washed three times with D-PBS, and stained with secondary antibody (Alexa Fluor 568, goat anti mouse, Thermo Fisher A11004 and Alexa Fluor 488, goat anti-rabbit Thermo Fisher A11034) 1:400 in 0.02% Triton X-100 in D-PBS for 1 h. Cells were stored in the dark at 4°C until observed under the microscope.

### Microscopy

Images described on Fig EV1B are acquired on an EVOS FL Microscope (Thermo Fisher) equipped with a GFP LED light cube. Images described on Fig EV1C are acquired on a Zeiss Ax10 Observer D1 fluorescence microscope system equipped with AxioCam and an HXP 120V lamp, with an Objective Plan-Apochromat 20× and ZEN acquisition software.

All other fluorescent images are acquired on a Nikon Ti-E automated epifluorescent microscope. The microscope was equipped with a DS-Qi2 camera and a Lumencor Sola SE II 365 LED lamp. The filters sets were provided by Nikon for DAPI (DAPI-5060C), Alexa Fluor 488 (FITC-3540C), and Alexa Fluor 568 (mCherry C M343564, Sembrock)). The objectives used were Apo 20× Lambda, numerical aperture 0.75 (Nikon). We used a hardware-based autofocusing system called the perfect focus system (PFS) from Nikon to automatically focus the cells in the field of view.

### Image analysis

For quantification of Ki67 and Polr2a, images were analyzed by CellProfiler with custom-made programs. The nuclei were segmented from DAPI channel image by automated thresholding and watershed procedure to split touching nuclei. Based on the segmented nuclei in each image, the mean intensities of the nuclear Ki67 or POLR2A were measured. For quantification of CCNA2 experiments, the nuclei were segmented as described above in

ImageJ and the areas of the individual segmented nuclei were measured and plotted.

GOLGA2 levels were automatically measured by specially developed KNIME workflow. First, the nuclei were segmented from DAPI channel image by applying median filtration for noise suppression, automated thresholding, and watershed procedure to split touching nuclei. Cellular regions were defined by dilating nuclear masks. To segment of GOLGA2 structures in Alexa 568 channel, top-hat filter was applied to remove local background followed by automated thresholding. Total intensity of segmented fragments in Alexa 568 channel was measured for each cell and corrected for background value estimated from image regions without cells. Additional measurements for individual cells in DAPI and in Alexa 568 channels were used to perform follow-up quality control of measurements using shinyHTM software (https://github.com/embl-cba/shinyHTM). Mitotic cells, out-of-focus cells, and image regions with unspecific antibody stainings were excluded from the analysis.

### Flow cytometry

For isolation of stable Cas9-GFP-expressing cell populations, cells were sorted using FACSAria™ II (Becton Dickinson). Cell populations were gated on a forward (FSC)/side scatter (SSC) plot. Cells were further gated on forward-area (FSC-A)/forward-height scatter (FSC-H) plot to determine single cells. Single cells were further gated on side-area scatter (SSC-A)/(405-450/50A) to determine living cells based on Sytox Blue (Molecular Probes™ S34857, 1 µM final concentration) dead cell staining. Live cells were further gated to determine eGFP (488-530/30-A) cell populations and sorted using a 45psi/85µm Nozzle.

For the analysis of Ki67 antibody staining, cells were fixed and permeabilized directly on 96-well plates using the cell fixation and permeabilization kit (Abcam). Then, the Ki67 antibody was incubated with the samples (1:400) overnight at 4°C followed by the incubation of the secondary antibody coupled to Alexa-488 (Alexa Fluor 488, goat anti-rabbit Thermo Fischer A11034, 1:1,000) for 1 h at room temperature. Samples were then analyzed on a FACS LSRFortessa™ mounted on high-throughput samples (HTS) (BD Biosciences, USA).

### Biological samples/patients

Human tissue samples were obtained from lung resections of Caucasian patients that had not received chemotherapy or radiation before surgery at the Thorax Clinic (Heidelberg, Germany). The protocol for tissue collection was approved by the ethics committees of the University of Heidelberg and the Ludwig-Maximilians-Universität München (reference S-270/2001 333-10 and 17-166). Patients gave informed consent and remained anonymous in the context of this study.

### Fibroblast isolation and expansion

Primary normal human lung fibroblasts (HLF) were isolated by explant outgrowth from distal airway-free lung tissue obtained from patients undergoing lung resection due to primary squamous cell carcinomas (SCC) at the Thoraxklinik (Heidelberg, Germany) (Woo *et al*, 2015; Tiran *et al*, 2017). The explanted tissue was tumor-free and did not present emphysema, fibrosis, or any inflammatory changes as shown by histopathology. Moreover, the patients displayed normal lung function, as determined by spirometry. Cells were collected from several explant pieces and maintained in DMEM supplemented with glutamine, penicillin–streptomycin, sodium

pyruvate, nonessential amino acids, and 2% FBS. Cells were used within passages 2–4.

### Primary tumor cell isolation and culture

Tumor cells were obtained by mechanical and enzymatic dissociation of dissected squamous cell carcinomas and adenocarcinomas as follows. Freshly obtained tumors from patients treated at Thoraxklinik (Heidelberg, Germany) were transported in $CO_2$-independent medium supplemented with 1% BSA and amphotericin, washed with HBSS, and minced with sterile razor blade and eye scissors. Tumor pieces were dissociated into single cell suspensions with the human tumor dissociation kit following the manufacturer's instructions (Miltenyi Biotec, Germany). Enzymatic reaction was stopped by adding 10% FBS, and single cells were collected by sequential filtering through cell strainers of 100, 70, and 40 μm (BD Falcon). Cells were centrifuged, resuspended in ACK lysis buffer (Thermo Fisher), and incubated 3 min at room temperature to lyse erythrocytes. After two washes with HBSS, cells were resuspended in SAGM (Lonza) supplemented with 1% FBS. Tumor cells were obtained by differential seeding and trypsinization and used within passages 1–3.

### Calculation of LOD Scores

We calculated LOD scores for the analysis of the oncogene screens. In these screens, the LOD score is equal to the logarithm to base 10 of the ratio of the probability that the effect of the gRNA is linked to the effects of POLR2A gRNA(our positive control that causes lethality when knocked out) to the effect of the gRNA are linked to the effect of scrambled gRNA (our negative control). Higher LOD scores are an indication of a gene KO behaving similarly to POLR2A KO, thus causing loss of cell viability. These probabilities are calculated based on the assumption that the viability values derived from both negative and positive controls are normally distributed. In terms of significance, a LOD score of 3 means the odds are a thousand to one that the two phenotypes are linked, and this is used as the traditional LOD score critical value. Using this scoring system, we can identify gene KOs that are more likely to cause loss of cell viability across different cancer cell lines and look for similarities and differences between different cell lines. There are two important parameters to determine the quality of a screen: (i) distribution of controls and (ii) the penetrance of the phenotypes. Higher LOD scores can be best obtained when the distribution of positive and negative controls is narrow and well separated. This also serves as a quality control in our screens as wide and intersecting distributions will not yield high LOD scores.

### Statistical analysis

Statistical tests were performed using Wilcoxon–Mann–Whitney $U$ test unless stated otherwise. The number of the independent replicates is stated on the figure legends or in the Materials and Methods Section.

## Data availability

Representative raw images of the microscopy-based analyses described in this study are deposited to BioStudies with the accession number: S-BSST309.

Expanded View for this article is available online.

### Acknowledgements

We are grateful to Sebastian M. Waszak for the conceptualization of the LOD score-based analysis of the screens. We also thank Christian Tischer for help with CellProfiler and Christa Stolp for help with collecting primary material. Samples were provided by Lung Biobank (Heidelberg, Germany)—a member of the Biomaterial bank Heidelberg (BMBH), the tissue bank of the National Center for Tumor Diseases (NCT), and the Biobank platform of the German Center for Lung Research (DZL). We thank Jan Mauer and Anton Khmelinskii for critical comments on the manuscript. This study was supported by Merck KGaA, Darmstadt, Germany.

### Author contributions

BRM conceived of and led the project. BN established the initial solid-phase transfection protocol, and BRM, BN, and ÖS optimized the conditions. ÖS did the majority of the experiments with help from NB, PR, SP, and AB. SR performed and analyzed imaging experiments. AH performed image analysis for GOLGA2 staining. ML, RJ and HW and TM isolated and provided human primary material tissue. FTZ and AA provided critical input and materials. KR generated the RPE-1 $TP53^{-/-Cas9}$ cells. BRM wrote the paper with significant contributions from ÖS and SR and with input from all authors.

### Conflict of interest

Frank T. Zenke is an employee of Merck KGaA, Darmstadt, Germany. Özdemirhan Serçin, Paris Roidos, Nadja Ballin, Spyridon Palikyras, Anna Baginska, Katrin Rein, Maria Llamazares, Renata Z. Jurkowska, and Balca R. Mardin are employees of BioMed X Innovation Center (GmbH), Heidelberg, Germany. Amir Abdollahi is academic mentor @ BioMedX and receives research grants from Merck KGaA. The other authors declare no conflict or financial interests.

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
