## [Review Process File · Molecular Systems Biology]

A solid-phase transfection platform for arrayed CRISPR screens

Özdemirhan Serçin, Sabine Reither, Paris Roidos, Nadja Ballin, Spyridon Palikyras, Anna Baginska, Katrin Rein, Maria Llamazares, Aliaksandr Halavatyi, Hauke Winter, Thomas Muley, Renata Jurkowska, Amir Abdollahi, Frank T. Zenke, Beate Neumann, Balca R. Mardin.

Review timeline:

Submission date:	8 th May 2019
Editorial Decision:	19 th June 2019
Revision received:	30 th October 2019
Editorial Decision:	21 st November 2019
Revision received:	24 th November 2019
Accepted:	27 th November 2019

Editor: Maria Polychronidou

Transaction Report:

1st Editorial Decision

19th June 2019

Thank you again for submitting your work to Molecular Systems Biology. We have now heard back from the three referees who agreed to evaluate your study. As you will see below, the reviewers acknowledge that the presented method seems potentially useful for the field. They raise however a series of concerns, which we would ask you to address in a major revision.

I think that the reviewers' recommendations are rather clear and there is therefore no need to repeat the comments listed below. Some of the more fundamental points are raised by reviewer #3 and refer to the need to provide further experimental support for the suitability of the method for phenotypic screens. Moreover, reviewer #2 mentions that some discussion should be included to better contextualize the method and to mention potential limitations.

All other issues raised by the reviewers need to be satisfactorily addressed. Please feel free to contact me in case you would like to discuss in further detail any of the issues raised by the reviewers.

REFERE REPORTS

Reviewer #1:

In this paper, Serçin and colleagues develop an approach for solid phase transfection of CRISPR/Cas9 reagents. They demonstrate delivery of both gRNA and gRNA/Cas9 RNPs into cell lines and primary cells, and effective gene editing in response. It is a direct, nicely written paper documenting a useful advance.

I enjoyed the paper, and suspect the main criteria for suitability of publication will be novelty and impact. The technique has multiple advantages (page 7, line 17-27): arraying reagents without batch effects onto many plates, storing them over time, and seeding at low confluency. It is up to the Editor to decide how this fits with the scope of the journal.

Major comments:

- Likelihood model. The statistics are reasonably presented in the form of log likelihood ratios. The core part of this calculation is the likelihood function - please specify what was used. Did you use empirical distribution of controls - in this case, how do you deal with values outside their range? Did you use a Gaussian fit - in this case, did you check for normality of the values?

Minor comments:

- terminology: guide RNA: tracrRNA complexes are referred to. Usually, gRNA refers to a single molecule RNA that is a fusion of tracrRNA and crRNA; the gRNA:tracrRNA complex is puzzling.
- the statistical methodology writeup in methods and main text (e.g. lines 31-33 on page 6) is not precise, please re-consult the statistics expert to clarify.
- the Cas9 lines used a dox-inducible system. Please clarify how the Cas9 induction was performed - I could not find the details in Methods.
- Figure 1B - were all cells fixed every 24 hours as it reads in the legend?
- Figure 3A - perhaps distinguish the two different types of cells in the figure, e.g. by marker type

Reviewer #2:

Sercin et al describe an alternative approach to arrayed CRISPR screening, using transfection in solid phase to deliver gRNA or RNP payloads to cells. The approach uses pre-treated transfection plates, allows growth of cells directly on the transfection substrate, and enables medium-throughput microscopy screening of individual perturbagens.

Issues:

The technical achievements described in this manuscript are adequate. My main concern with this study is that it does not sufficiently describe the context into which these achievements fit. There is a substantial body of work preceding this paper, which describes similar methods with pre-CRISPR perturbagens. There is also a fair bit of work describing the features and limitations of CRISPR/Cas9 systems which should be addressed.

There is an extensive literature on cell microarrays from the early to mid 2000s, starting (I believe) with Ziauddin & Sabatini, Nature 2001, wherein transfection mix is printed on a glass slide and cells are grown in a lawn over it, with the goal of overexpressing cDNA. This technology was later adapted to siRNA/shRNA and, in conjunction with automated microscopy, enabled high throughput, high-content screening for complex phenotypes. Sercin et al should at least acknowledge this prior work and discuss their contribution in context. Of course CRISPR is a game-changer and the ability to do similar screens, even in a microwell-based format, is an advance, but the limitations should be discussed. What are the constraints of this system relative to cell microarrays? It seems to require the transfection of cells at lower density and longer assay times, consistent with the genetic (vs. transcriptomic) perturbation of CRISPR vs. RNAi.

Furthermore, the idiosyncrasies of the CRISPR system should be addressed. It is disappointing that the authors used a scrambled gRNA control, since it is well known that CRISPR/Cas9 causes locus-nonspecific fitness defects that will affect the signal to noise of the entire assay. These locus-nonspecific effects should be disambiguated from true signal by using gRNA that target known nonessential genes or intergenic regions. Moreover, CRISPR/Cas9 targeting copy-amplified regions causes cellular toxicity; an ERBB2 "positive control" in NCI-N87 cells is almost certainly copy amplified. It is difficult to disambiguate ERBB2 locus-specific from copy amplified locus-nonspecific effects in these cells.

The complete set of control experiments to address these idiosyncrasies need not be completed to publish this paper, but a discussion of these features/limitations is absolutely necessary. This is not a marketing brochure. The technology is potentially quite interesting but labs who adopt this approach should be armed with as much knowledge as possible in order to make an informed choice.

Reviewer #3:

Özdemirhan Serçin et al. investigate the new strategies for CRISPR-based forward genetic screen. They coat microwells with gRNA:tracrRNA complexes and the transfection reagent, and demonstrate efficient sgRNA-guided knockout in multiple seeded Cas9-expressing cell lines. They also show that the solid-phase transfection platform has high-efficiency of knock-out using Cas9:gRNA ribonucleoprotein (RNP) to cancer cell line and lung primary epithelial cells. This is a straightforward method that has the potential to be applied to phenotypic screens.

In general, the results and conclusions are convincing. Technically they established a new platform for reverse transfection of gRNAs in many different cell lines. The transfection complex and the synthetic gRNA complex or the Cas9-RNP complex can be easily coated on plate and efficiently delivered.

This method uses lipid transfection, reducing biases and labor from lentivirus based methods. It also shows significantly lower cytotoxicity comparing to the previous lipid-transfection based Crispr strategy. The solid-phase platform increases the phenotype readouts, including imaging, sequencing and cell viability assay. The manuscript would need some more experiments in order to increase the future usage of the platform in other laboratory.

1. The authors use only guides for 3 genes and as a readout mainly use viability. The strongest point of the paper is to use such platform for phenotypic screens. I believe the authors should test more extensively the platform looking at multiple genes not only in viability assays (e.g. cell shape, cell size, antibody staining etc)

2. The authors show the percentage of viable cells however would be better if they could show how many cells have an efficient knockout. This could be done in the Golgi assay, quantifying number of cells with loss of staining. And this should be done with multiple genes and staining to show what is the general efficiency of knock out of different genes in a population.

3. Did the author think of a way to distinguish single cells that had the guide delivered or not in a population of cells? This could be an interesting addition.

4. In Fig 1E, it would be informative to demonstrate the performance of solid-phase platform with high confluency cell culture.

5. For the control of the phenotype readouts in different cell lines, would be helpful to add the sample without scrambled RNP or sgRNA delivery, to give information on the nature of the cell lines. This could help to understand the different phenotype penetrance between cell lines.

6. Fig EV3E, Polr2a gRNA delivered cells show less POLR2A staining signal as well as less cell viability. It would be more informative to show the quantification of immunostaining intensity per nucleus.

7. Page4 line 7 - I suppose the second 'either' should be removed.

8. Page4 line 18 - "to PLK1 knockdown by siRNA (Fig EV1A, B)." - I could not find the data with siRNA.

9. Page4 line 22 - "after transfection (Fig 3C, D)." - I suppose it would be (Fig EV3C, D)

Response to Reviewer Comments

We would like to thank all the reviewers for their valuable comments on our manuscript. We were delighted to see that the reviewers were overall positive about our method, acknowledging the potential importance and novelty of the approach. They raised some important points and we put substantial effort into addressing all of the concerns by clarifying several specific aspects as well as by providing additional data and analyses in our revised manuscript. Please find below our point-by-point response to reviewers' comments.

Reviewer #1:

In this paper, Sercin and colleagues develop an approach for solid phase transfection of CRISPR/Cas9 reagents. They demonstrate delivery of both gRNA and gRNA/Cas9 RNPs into cell lines and primary cells, and effective gene editing in response. It is a direct, nicely written paper documenting a useful advance.

I enjoyed the paper, and suspect the main criteria for suitability of publication will be novelty and impact. The technique has multiple advantages (page 7, line 17-27): arraying reagents without batch effects onto many plates, storing them over time, and seeding at low confluency. It is up to the Editor to decide how this fits with the scope of the journal.

We thank the reviewer for the positive comments.

Major comments:

Likelihood model. The statistics are reasonably presented in the form of log likelihood ratios. The core part of this calculation is the likelihood function - please specify what was used. Did you use empirical distribution of controls - in this case, how do you deal with values outside their range? Did you use a Gaussian fit - in this case, did you check for normality of the values?

In the revised manuscript, following the reviewer's suggestion we provide more detailed explanation of the functions used in the LOD score calculation. The updated version can be found in the **Figure EV6A-C** of our revised manuscript.

In particular, the control distributions in our screens are based on real data derived from each plate, therefore there are no values outside the range.

We tested for the normality of the Scrambled and *POLR2A* gRNA distributions. We plotted the distributions of both control gRNAs and applied Kolmogorov-Smirnov test with a normally distributed sample. In both cases, we have not observed any significant difference from the normal distribution so for this reason we assume

normality of these control samples. We included the distribution of the control guides as well as the qqplots in the Figure **EV6C** in the revised manuscript.

Minor comments:

- terminology: guide RNA: tracrRNA complexes are referred to. Usually, gRNA refers to a single molecule RNA that is a fusion of tracrRNA and crRNA; the gRNA:tracrRNA complex is puzzling.

Throughout the revised manuscript, we corrected this and now stick to the terminology as suggested by the reviewer.

- the statistical methodology writeup in methods and main text (e.g. lines 31-33 on page 6) is not precise, please re-consult the statistics expert to clarify.

We integrated more information in the main text (**page 8, lines 11-22**) and in the material and method section (**page 18, calculation of LOD scores**) as well as in the Figure EV6A-C on how we calculated these scores.

- the Cas9 lines used a dox-inducible system. Please clarify how the Cas9 induction was performed - I could not find the details in Methods.

The method of Cas9 induction (whenever applicable) is included for each cell line in the table we now provide as **Table EV1** (Cell Lines and Media) in our revised manuscript.

- Figure 1B - were all cells fixed every 24 hours as it reads in the legend?

Indeed, the cells were fixed after 24, 48 and 72 hours of transfection. To make this section clear, we have updated the figure legend (**Figure 2A** of the revised manuscript).

- Figure 3A - perhaps distinguish the two different types of cells in the figure, e.g. by marker type

Following this suggestion, we have updated the former figure 3A (**Figure 4A** in the revised manuscript).

Reviewer #2:

Sercin et al describe an alternative approach to arrayed CRISPR screening, using transfection in solid phase to deliver gRNA or RNP payloads to cells. The approach uses pre-treated transfection plates, allows growth of cells directly on the transfection substrate, and enables medium-throughput microscopy screening of individual perturbagens.

Issues:

The technical achievements described in this manuscript are adequate. My main concern with this study is that it does not sufficiently describe the context into which these achievements fit. There is a substantial body of work preceding this paper, which describes similar methods with pre-CRISPR perturbagens. There is also a fair

bit of work describing the features and limitations of CRISPR/Cas9 systems which should be addressed.

There is an extensive literature on cell microarrays from the early to mid 2000s, starting (I believe) with Ziauddin & Sabatini, Nature 2001, wherein transfection mix is printed on a glass slide and cells are grown in a lawn over it, with the goal of overexpressing cDNA. This technology was later adapted to siRNA/shRNA and, in conjunction with automated microscopy, enabled high throughput, high-content screening for complex phenotypes. Sercin et al should at least acknowledge this prior work and discuss their contribution in context. Of course CRISPR is a game-changer and the ability to do similar screens, even in a microwell-based format, is an advance, but the limitations should be discussed. What are the constraints of this system relative to cell microarrays? It seems to require the transfection of cells at lower density and longer assay times, consistent with the genetic (vs. transcriptomic) perturbation of CRISPR vs. RNAi.

We thank the reviewer for acknowledging the technical strength of our manuscript. We also thank this reviewer for pointing out the oversights in our manuscript, especially with regards to missing literature and discussion. We have made substantial effort in extending both the introduction and discussion of our revised manuscript, accommodating all the changes suggested by the reviewer. For instance, in the introduction we discussed the prior work that was done with cDNAs and siRNAs that we regrettably did not discuss in the previous version of our manuscript (**page 3, line 24 - page 4, line 8**). In addition, we also discussed the potential limitations of the solid phase transfection platform in combination with the gRNA screens (**page 9, line 31 – page 10, line 13**).

Furthermore, the idiosyncrasies of the CRISPR system should be addressed. It is disappointing that the authors used a scrambled gRNA control, since it is well known that CRISPR/Cas9 causes locus-nonspecific fitness defects that will affect the signal to noise of the entire assay. These locus-nonspecific effects should be disambiguated from true signal by using gRNA that target known nonessential genes or intergenic regions.

There are several examples in the literature that use non-targeting negative controls the sequence of which cannot be found on the host's genome to calculate effects that can be caused by potential off-target effects. One of the recent studies analyzed the effects of 5,644 non-targeting guides and 6,750 'safe-targeting' guides (guides that target genomic sites with no annotated function) on different types of CRISPR screens. While safe targeting guides may have some other advantages in the positive selection screens, in the growth & viability screens, they were depleted more than the non-targeting guides, suggesting that safe-targeting guides are more toxic for the cells than non-targeting guides (Morgens, David W., et al. Nature Communications, 2017, PMID: 28474669). Following this rationale, we also designed a non-targeting scrambled gRNA that we use throughout the manuscript. However, we understand that this in principle may cause some unintended fitness effects. In order to answer the reviewer's comment and to demonstrate that negative controls used in our experiments do not cause major differences in fitness, we compared our non-targeting negative control gRNA side by side to the effects of several other negative controls. In addition to our scrambled control, we now used 3 more control gRNAs:

an additional non targeting gRNA, a gRNA targeting an intronic region and a gRNA targeting a nonessential gene (HPRT). In all cases, in four different cell lines (NCI-

H358, NCI-N87, RPE-1 and HEK293T) we have not observed major changes in cell viability. These data are now provided in **Figure EV3C**.

Moreover, CRISPR/Cas9 targeting copy-amplified regions causes cellular toxicity; an ERBB2 "positive control" in NCI-N87 cells is almost certainly copy amplified. It is difficult to disambiguate ERBB2 locus-specific from copy amplified locus-nonspecific effects in these cells.

It is indeed true that ERBB2 is copy amplified and it is now well acknowledged that high copy numbers affect the viability of the cells when targeting these regions by gRNAs and ERBB2 dependency may be confounded by this effect. However, in our screens, dependency on KRAS does stem from an activating mutation, thus is directly a measure of the oncogenic activity of KRAS that the cells rely on. For this reason, we still think that the oncogene addiction screens are a valuable addition to the manuscript, in terms of demonstrating how different genotypes can be specifically targeted and how this can be assessed. However, in the revised manuscript, we now carefully discuss the effects of targeting genes on copy number amplified regions and include the references (**page 8, lines 2-10**).

The complete set of control experiments to address these idiosyncrasies need not be completed to publish this paper, but a discussion of these features/limitations is absolutely necessary. This is not a marketing brochure. The technology is potentially quite interesting but labs who adopt this approach should be armed with as much knowledge as possible in order to make an informed choice.

We thank the reviewer for this thoughtful concern and we agree that the limitations of the method should be discussed to allow the users to make the most informed decision about any screens. To this end, we now address the potential limitations as well as advantages of our method. Throughout our revised manuscript, and especially in the final paragraphs of the manuscript (**pages 9-10**) we now comment on:

- Although the equipment is not sophisticated, solid phase transfection still needs extra equipment (vacuum centrifuge) for lyophilization of the transfection mixes.
- While the immortalized or transformed cell lines maybe efficiently transfected with this method, the primary cells may need more optimization to achieve highest efficiency. We hope to provide a first step towards this goal.
- The solid phase transfection platform does not work on suspension cells.

As a further discussion point, we analyzed how many genes can in principle be targeted by solid phase transfection. For this we utilized the study by McShane et al (Cell, 2016, PMID: 27720452) that measures the protein turnover rates in RPE-1 cells. Assuming most of the experiments should be done in 3-5 days post transfection, we wondered how many proteins can be efficiently targeted by CRISPR/Cas9 based methods such as solid phase transfection. Using the dataset from McShane et al., we considered 1-state degradation model for exponentially degraded proteins (ED) and 2-state degradation model for non-exponentially degraded (NED) proteins. We then plotted the density of all the proteins that could be identified in this study and classified the proteins as "targetable" and "hard to target" in CRISPR/Cas9 based arrayed screens. To this end, we used the following thresholds:

- 1- Proteins with half life of <120

2- Proteins with half-life of ≥ 120 h

Based on these thresholds, we see that at least in RPE-1 cells 17.7% of the proteins cannot be efficiently targeted. We present the results of this analysis in **Figure EV5D** and discuss all these limitations in the revised manuscript (**page 10, lines 1-13**).

Reviewer #3:

Özdemirhan Serçin et al. investigate the new strategies for CRISPR-based forward genetic screen. They coat microwells with gRNA:tracrRNA complexes and the transfection reagent, and demonstrate efficient sgRNA-guided knockout in multiple seeded Cas9-expressing cell lines. They also show that the solid-phase transfection platform has high-efficiency of knock-out using Cas9:gRNA ribonucleoprotein (RNP) to cancer cell line and lung primary epithelial cells. This is a straightforward method that has the potential to be applied to phenotypic screens. In general, the results and conclusions are convincing. Technically they established a new platform for reverse transfection of gRNAs in many different cell lines. The transfection complex and the synthetic gRNA complex or the Cas9-RNP complex can be easily coated on plate and efficiently delivery. This method uses lipid transfection, reducing biases and labor from lentivirus based methods. It also shows significantly lower cytotoxicity comparing to the previous lipid-transfection based Crispr strategy. The solid-phase platform increases the phenotype readouts, including imaging, sequencing and cell viability assay. The manuscript would need some more experiments in order to increase the future usage of the platform in other laboratory.

We thank the reviewer for acknowledging the quality of the data as well as the potential of our method to be used in screens with gRNA/RNPs. We also thank this reviewer for suggesting several experiments that strengthen our manuscript. We have made major improvements to the manuscripts based on the suggestions of the reviewer and now present data with more marker genes that showcase the applicability of the platform to phenotypic screens.

1. The authors use only guides for 3 genes and as a readout mainly use viability. The strongest point of the paper is to use such platform for phenotypic screens. I believe the authors should test more extensively the platform looking at multiple genes not only in viability assays (e.g. cell shape, cell size, antibody staining etc)

Following the reviewer's suggestion, in the revised manuscript, we present that from 4 genes that exhibit clear phenotypes upon disruption that can be followed by imaging-based experiments, assessing either the changes in the nuclear morphology and size; *PLK1*, *CCNA2* or measuring the loss of signal after antibody staining; *GOLGA2* and *MKI67*.

In all cases, we demonstrate efficient knock down of all these genes either based on morphological changes or based on loss of staining. All these experiments are now presented in the main **Figures 2 (for gRNA) and 3 (for RNP)** of the revised manuscript, and described extensively in the main text (**page 4 line 30 – page 5,**

line 18), making the strong point that the solid phase transfection platform can also be used for phenotypic screens.

2. The authors show the percentage of viable cells however would be better if they could show how many cells have an efficient knockout. This could be done in the Golgi assay, quantifying number of cells with loss of staining. And this should be done with multiple genes and staining to show what is the general efficiency of knock out of different genes in a population.

In our revised manuscript, as described in the point above, we have performed experiments targeting *GOLGA2* and *MKI67* in combination with antibody staining. In both cases we quantified the signals in Golgi and nuclei, respectively. These data are now provided in **Figures 2D-G** for targeting by gRNA and **Figure 3C, D** for targeting by RNP. We believe that these experiments demonstrate the efficiency of solid phase transfection as assessed by the level of proteins left upon transfection by antibody staining.

3. Did the author though of a way to distinguish single cells that had the guide delivered or not in a population of cells? This could be an interesting addition.

To our knowledge, the only way to easily visualize the gRNA entering the cells is via using a tracrRNA that is attached to a fluorescent dye (e.g. Atto550). However, this does not ensure that the editing has taken place. In the past, we have tried several times to use the tracr-Atto550, however unfortunately failed to detect a direct correlation between the red stained cells and the phenotypic penetrance. One potential issue here stems from the fact that Atto550 is fluorescent only after 24 hours of transfection and most phenotypes after editing are visible only after 72 hours.

For instance, in the figure below that we included here for the reviewer, we transfected cells with *Plk1* crRNA and tracrRNA-atto550. While we see that the majority of the cells have the red dye indicating a successful transfection, not all the cells show the phenotype that is associated with *Plk1* downregulation (Prometaphase arrest).

While we agree with the reviewer that it would be really helpful to identify cells that have the gRNA and in principle this is possible with the tracrRNA-atto550, since we cannot make a direct correlation between the cells that are transfected and the cell that are edited that will show a phenotype, we refrain from making conclusions based on such experiments.

Figure: RPE-1 cells are transfected with the indicated crRNAs with tracrRNA coupled to a fluorescent dye, Atto550. 24 hours post transfection, the cells were fixed and analyzed by microscopy. After 24 hours, while almost all the cells appear to be red which is an indicative of high efficiency of transfection, only approximately 10% of cells showed signs of prometaphase arrest in cells transfected with *PLK1* gRNA. Thus, we concluded that it is currently difficult to estimate the % of edited cells based on the % of transfected cells.

4. In Fig 1E, it would be informative to demonstrate the performance of solid-phase platform with high confluency cell culture.

In order to address this comment from the reviewer we performed solid phase transfection with high confluency cell culture (20.000 cells/well at the time of transfection) in four different cell lines (NCI-H358, NCI-N87, RPE-1 and HEK293T). In H358 cells, we have not observed a dramatic effect on efficiency as assessed by cell viability measurements upon *POLR2A* gRNA transfection when high numbers of cells were seeded. In N87 and in HEK293T cells, however, when 20.000 cells/well were seeded, we could no longer observe the lethal effect of *POLR2A* gRNA. In RPE-1 cells, some effect on viability could still be observed however the efficiency was rather low. The data is presented in **Figure 2H** of our revised manuscript. In conclusion, when high numbers of cells are seeded for solid phase transfection, the results can be variable and depend on each cell line, whereas when low numbers of cells are seeded in all cases, we achieved high efficiencies of transfection with high reproducibility. Altogether, these data are in line with our conclusions, suggesting that CRISPR screens to be carried out by solid phase transfection with low number of seeded cells in an efficient manner for extended periods of time without the need of additional handling steps. In the revised manuscript we point to these differences and discuss how different cell lines may respond differently to increased cell numbers in terms of efficiency in solid phase transfection (**page 6 lines 19-28**).

5. For the control of the phenotype readouts in different cell lines, would be helpful to add the sample without scrambled RNP or sgRNA delivery, to give information on the nature of the cell lines. This could help to understand the different phenotype penetrance between cell lines.

We provide two lines of evidence to show that the mock transfection does not cause major effects in different cell lines.

- (i) to demonstrate the general fitness of cells that are transfected with either no gRNA (mock) or scrambled gRNAs, we took transmission images of the NCI-H358, NCI-N87, RPE-1 and HEK293T cells 72 hours post transfection. In all four cell lines, the general fitness of the cells is not affected either by mock transfection or by transfection of scrambled gRNA. This data is now presented in **Figure EV3A** of our revised manuscript.
- (ii) For the different cancer cell lines that we used to test for viability, we have not observed major differences in mock transfection. These data are now provided in the **Figure EV3D**.

6. Fig EV3E, Polr2a gRNA delivered cells show less POLR2A staining signal as well as less cell viability. It would be more informative to show the quantification of immunostaining intensity per nucleus.

We think the way we presented this data may have caused some confusion since the quantification was in Figure 1C, while the corresponding images were presented in Figure EV3E. We apologize for this and now present the quantification of immunostaining intensity per nucleus together with the corresponding images in **Figure EV2H-I**.

7. Page4 line 7 - I suppose the second 'either' should be removed.

We corrected this mistake.

8. Page4 line 18 - "to PLK1 knockdown by siRNA (Fig EV1A, B)." - I could not find the data with siRNA.

The data is presented now in **Figure EV2A,B**.

9. Page4 line 22 - "after transfection (Fig 3C, D)." - I suppose it would be (Fig EV3C, D)

We corrected this mistake. The experiments with the Golga2 staining are now presented in **Figure 2D, E**.

2nd Editorial Decision

21st November 2019

Thank you for sending us your revised manuscript. We have now heard back from the reviewer who agreed to evaluate your study. As you will see below, reviewer #3 is satisfied with the modifications made. As such, I am glad to inform you that the study is now suitable for publication, pending some minor editorial issues.

REFEREE REPORTS

Reviewer #3:

The authors addressed all our concerns, especially adding more single cell analysis.

1, Yes, the authors present multiple phenotypes: staining for Ki67 for MKI67 gRNA and RNP, Golga2 staining for GOLAG2 gRNA and RNP, plot with nuclei size and nuclei density for PLK1 gRNA and RNP.

2-9 All the comments are addressed properly.

Corresponding Author Name: Balca R. Mardin

Manuscript Number: MSB-19-8983